# Branching-Bounded Contingent Planning via Belief Space Search

**Kevin McAreavey**[1]  and  **Kim Bauters**[1]  and  **Weiru Liu**[1]  and  **Jun Hong**[2]

[1]University of Bristol, UK

{kevin.mcareavey, kim.bauters, weiru.liu}@bristol.ac.uk

[2]University of the West of England, UK

jun.hong@uwe.ac.uk

## Abstract

A contingent plan can be encoded as a rooted graph where branching occurs due to sensing. In many applications it is desirable to limit this branching; either to reduce the complexity of the plan (e.g. for subsequent execution by a human), or because sensing itself is deemed to be too expensive. This leads to an established planning problem that we refer to as branching-bounded contingent planning. In this paper, we formalise solutions to such problems in the context of history-, and belief-based policies: under noisy sensing, these policies exhibit differing notions of sensor actions. We also propose a new algorithm, called BAO*, that is able to find optimal solutions via belief space search. This work subsumes both conformant and contingent planning frameworks, and represents the first practical treatment of branching-bounded contingent planning that is valid under partial observability.

## 1    Introduction

In planning under uncertainty, a contingent plan is the most general solution form. A typical encoding of a contingent plan is a rooted graph (or tree) that exhibits branching. This branching occurs because a contingent plan must account for all possible feedback that might be received during plan execution in response to *sensing*. However, in many applications it is desirable to limit this branching; either to reduce the complexity of the plan, or because sensing itself is deemed to be too expensive. This leads to an established planning problem (Baral, Kreinovich, and Trejo 2000; Meuleau and Smith 2002; Bonet 2010) that we will refer to as *branching-bounded contingent planning*.[1]

Our intuition is that there exists a positive correlation between the complexity of a contingent plan (e.g. how difficult it is for a human to comprehend or execute), and the amount of branching that it contains. Conversely, studies have shown that humans are demonstrably bad at following complex plans (Dodson et al. 2013). In this sense, branching-bounded contingent planning provides a means to ensure that plans are sufficiently simple so as to be understood by humans. This idea has previously been referred to as the "cognitive simplicity" of a plan (Meuleau and Smith

---

[1]Also known as *limited contingency planning* (Meuleau and Smith 2002). Not to be confused with other forms of planning with bounded parameters (e.g. see Section 4 for a discussion).

2002), and is an important consideration in numerous explainable AI planning (XAIP) applications, including where humans are required to verify plans generated by automated planners (Meuleau and Smith 2002), and where humans are required to execute such plans (Green et al. 2011).

Considerations around plan complexity also extend to the field of autonomous agents. For example, if agents have limited computational resources, then it may not be feasible to maintain the agent's belief state online, which precludes the direct use of functional plan representations such as belief-based policies (Kaelbling, Littman, and Cassandra 1998; Meuleau and Smith 2002). This is true of recent work on augmenting belief-desire-intention (BDI) agents with automated planners and reusable plans (Meneguzzi and De Silva 2015), where it is important to limit the complexity of new plans so as to maintain the agent's reactiveness: the greater the amount of branching in the plan, the greater the increase in the size of the agent's plan library, and the greater the computational cost associated with future plan selection.

A related approach to contingent planning is the field of conformant planning, which deals with domains that are non-observable (i.e. that have no sensing). Although such domains are sometimes dismissed as having little practical interest (Taig and Brafman 2015), a common motivation for conformant planning is in applications where sensing is deemed to be too expensive (Domshlak and Hoffmann 2006). This suggests that such applications are not truly non-observable, but rather that the use of sensing should be bounded (and potentially avoided altogether). In fact, branching-bounded contingent planning can be seen as a generalisation of both conformant and contingent planning.

The only practical treatment of branching-bounded contingent planning appears to be the work of Meuleau and Smith (2002) in the context of partially observable Markov decision processes (POMDPs). Informally, their method restricts sensing to a special *observe-and-branch* action, and then bounds the number of times that this action will be included in the solution plan. Unfortunately, the method is only valid under the assumption of full observability (Bonet 2010), and limits the generality of contingent planning by prohibiting richer forms of sensing. As far as we are aware, the only other work on bounded branching has been theoretical analyses of the complexity of various planning problems (Baral, Kreinovich, and Trejo 2000; Bonet 2010).

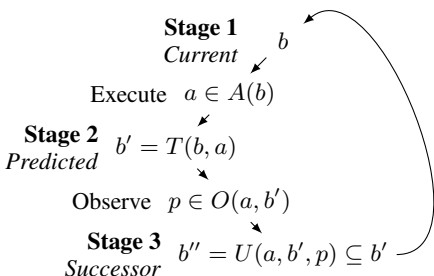

Figure 1: Belief state update procedure.

In this paper, we propose the first practical treatment of branching-bounded contingent planning that is valid under partial observability. We account for uncertainty over the initial state, action-effects, and observations. The main contributions are as follows: (i) we propose a definition of branching-bounded contingent plans in the context of history-based policies; (ii) we explore the implications of bounded branching in the context of belief-based policies; (iii) we propose a definition of branching-bounded contingent plans as generalised belief-based policies that track sensing; and (iv) we propose a variant of the AO* search algorithm for AND/OR graphs, called BAO*, that is able to find optimal solutions via belief space search. We will rely on the partially observable non-deterministic (POND) model of contingent planning, where a belief state is a set of states, but our ideas can likely extend to other models (e.g. goal-POMDPs, where a belief state is a probability distribution over the state space). We focus on offline planning (i.e. where a complete plan is generated and then executed in full) and will not consider online planning (i.e. where planning and plan execution are interleaved).

The remainder of the paper is organised as follows: in Section 2, we recall preliminaries on contingent planning and AND/OR graphs; in Section 3, we describe our solution to branching-bounded contingent planning; in Section 4, we discuss related work; and, in Section 5, we conclude.

## 2 Preliminaries

In this section, we recall necessary preliminaries on contingent planning and AND/OR graphs. We rely on some standard mathematical notation: $|S|$ is the cardinality of set $S$, $2^S$ is the powerset of $S$, $\mathbb{R}^+$ is the set of positive real numbers ($0 \notin \mathbb{R}^+$), and $\mathbb{N}$ is the set of natural numbers ($0 \in \mathbb{N}$).

### 2.1 Contingent Planning

A contingent planning domain is a tuple $(S, A, P, C, T, O)$ where $S$ is a set of states (called the state space) with $B = 2^S \setminus \{\emptyset\}$ the set of belief states (called the belief space), $A$ is a set of actions with $A(s) \subseteq A$ the set of applicable actions in state $s \in S$, $P$ is a set of percepts with $\varnothing_P \in P$ the null percept, $C : A \to \mathbb{R}^+$ is a cost function, $T : S \times A \to B$ is a transition function, and $O : A \times S \to 2^P \setminus \{\emptyset\}$ is an observation function. We say that $T$ (resp. $O$) is deterministic if $T(s, a)$ (resp. $O(a, s)$) is a singleton for each $s \in S$ and

each $a \in A$, otherwise $T$ (resp. $O$) is non-deterministic. We assume that a percept will be observed after every action-execution (see Figure 1), but if it is possible to observe *nothing* in state $s$, then this is encoded simply as the null percept $\varnothing_P \in O(s)$. Finally, a contingent planning task is a tuple $(M, b_1, S_G)$ where $M$ is a contingent planning domain, $b_1 \in B$ is an initial belief state, and $S_G \subseteq S$ is a goal. We say that a belief state $b \in B$ satisfies the goal iff $b \subseteq S_G$, i.e. when the agent is guaranteed to be in a goal state.

The notion of applicable actions is extended to a belief state $b \in B$ as $A(b) = A(s_1) \cap \cdots \cap A(s_n)$ such that $b = \{s_1, \ldots, s_n\}$, meaning that an action is applicable in $b$ iff it is applicable in each state $s \in b$.[2] The transition function $T$ is extended as a function $T : B \times A \to B$ defined as $T(b, a) = \{s \in T(s', a) \mid s' \in b\}$. The observation function $O$ is extended as a function $O : A \times B \to 2^P \setminus \{\emptyset\}$ defined as $O(a, b) = \{p \in O(a, s) \mid s \in b\}$. Finally, an update function $U : A \times B \times P \to B$ is defined as:

$$U(a, b, p) = \begin{cases} \{s \in b \mid p \in O(a, s)\} & \text{if non-empty} \\ \textit{undefined} & \text{otherwise} \end{cases}$$

Evidently, $U(a, b, p) = \textit{undefined}$ iff $p \notin O(a, b)$. We say that $b' = T(b, a)$ is a predicted belief state and $U(a, b', p) \subseteq b'$ is a successor belief state (again, see Figure 1).

In practice, contingent planning problems typically require a factorised representation in order to express problems of any meaningful complexity. For example, $S$ can be defined by a set of (independent) state variables, $A(s)$ (resp. $T$ and $O$) can be defined by a set of action schema preconditions (resp. effects and observations) associated with $A$, and $b_1$ (resp. $S_G$) can be defined by a logical formula over the set of state variables. We refer the interested reader to a planning language capable of expressing contingent planning problems, such as NuPDDL[3] or PO-PPDDL[4].

### 2.2 AND/OR Graphs

A (directed) graph is a tuple $(N, E)$ where $N$ is a set of nodes and $E \subseteq N \times N$ is a set of (directed) edges. A multigraph is a tuple $(N, I, E')$ where $I$ is a set of identifiers and $E' \subseteq N \times I \times N$ is a set of multiedges such that for each $i \in I$ we have that $(N, \{(n, n') \mid (n, i, n') \in E'\})$ is a graph. Given nodes $n, n' \in N$ in a multigraph, then $n$ is said to be a parent of $n'$ (resp. $n'$ is a child of $n$) iff $(n, i, n') \in E'$ for some $i \in I$. A node $n \in N$ is said to be a branch point if $n$ has more than one child. Given nodes $n_1, n_{m+1} \in N$ in a multigraph, then a sequence of nodes and identifiers $(n_1, i_1, \ldots, n_m, i_m, n_{m+1})$ is a path from $n_1$ to $n_{m+1}$ iff each $n_j$ is unique and $(n_j, i_j, n_{j+1}) \in E'$ for $j = 1, \ldots, m$. A multigraph is acyclic if, for each $n \in N$, there does not exist a path from $n$ to itself. A rooted (and connected) multigraph is a tuple $(N, I, E', n)$ where $(N, I, E')$ is a multigraph and $n \in N$ is a root node such that, for each $n' \in N$, there exists a path from $n$ to $n'$.

---

[2]This is the cautious approach to applicable actions; its dual $A(s_1) \cup \cdots \cup A(s_n)$ is possible but complicates later definitions.

[3]http://mbp.fbk.eu/NuPDDL.html

[4]http://users.cecs.anu.edu.au/~ssanner/IPPC_2011

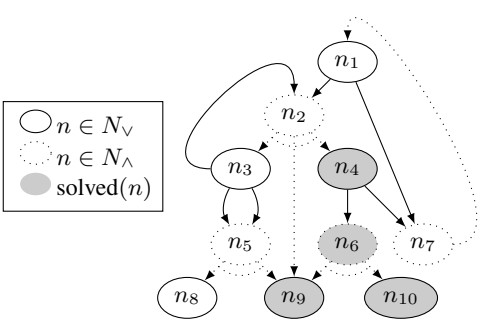

Figure 2: AND/OR graph.

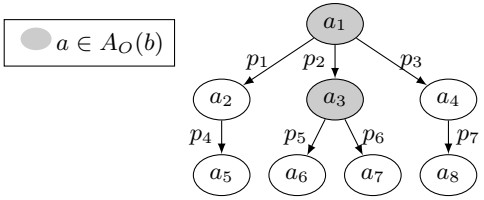

Figure 3: History-based $k$-branching-bounded plan, $k \geq 2$.

An AND/OR graph $(N_\vee \cup N_\wedge, I_\vee \cup I_\wedge, E_\vee \cup E_\wedge, n)$ is a rooted multigraph where $N_\vee$ (resp. $N_\wedge$) is a set of OR-nodes (resp. AND-nodes), $I_\vee$ (resp. $I_\wedge$) is a set of OR-identifiers (resp. AND-identifiers), $E_\vee \subseteq N_\vee \times I_\vee \times N_\wedge$ (resp. $E_\wedge \subseteq N_\wedge \times I_\wedge \times N_\vee$) is a set of OR-edges (resp. AND-edges), and $n \in N_\vee$ is a root node. An AND/OR graph is typically used to reduce problems into decomposable sub-problems. Intuitively, an AND-node is a solution if each of its child nodes is a solution, while an OR-node is a solution if it is a primitive solution, or at least one of its child nodes is a solution (e.g. see Figure 2). In the next section, we will demonstrate how branching-bounded contingent planning can be cast as a search problem over AND/OR graphs.

## 3 Framework

In this section, we formalise branching-bounded contingent planning in the context of history-, and belief-based policies, and propose a new solution that relies on belief space search.

### 3.1 History-Based Policies

An execution is a possibly infinite sequence $(a_1, p_1, a_2, p_2, \dots)$ where $a_i \in A$ and $p_i \in P$. A finite execution is also called a history, with $H$ the set of histories (called the history space). The length of history $h_{i+1} = (a_1, p_1, \dots, a_i, p_i)$ is defined as $|h_{i+1}| = i$. A history-based policy is a function $\pi_h : H' \to A$ where $H' \subseteq H$. The executions that are possible with respect to $b_1$ and $\pi_h$ are defined inductively along with their associated belief states as follows: the empty execution $h_1$ is possible and $b_1$ is its belief state; if $h_i$ is possible and $b_i$ is its belief state, then $h_{i+1} = (h_i, a_i, p_i)$ is possible and $b_{i+1} = U(a_i, b'_i, p_i)$ is its belief state iff $h_i \in H'$ such that $a_i = \pi_h(h_i)$, $p_i \in O(a_i, b'_i)$, and $b'_i = T(b_i, a_i)$. An execution $h$ that is possible with respect to $b_1$ and $\pi_h$ is said to be complete if $h \notin H'$ or if $h$ is infinite. Finally, $\pi_h$ is said to be a *strong* solution to a contingent planning task $(M, b_1, S_G)$ if each complete execution $h_i$ is finite and $b_i \subseteq S_G$.

**Definition 1.** *A history-based contingent plan is a history-based policy $\pi_h$ that is a strong solution to a contingent planning task $(M, b_1, S_G)$.*

A history-based policy $\pi_h$ can be encoded as a rooted acyclic multigraph over histories (called a history-based pol-

icy graph) where $\pi_h$ is a node label function and where each multiedge identifier is a percept (Kaelbling, Littman, and Cassandra 1998).[5] More precisely, a history-based policy graph is a rooted tree. To execute the plan, an agent simply needs to execute the actions specified by node labels, while tracing a single path in line with observed percepts. In this way, branch points are those nodes where no single percept is guaranteed to occur after executing the specified action, while the actions themselves can be thought of as information gathering actions, called *sensor* actions.

**Definition 2.** *The set of history-based sensor actions in belief state $b \in B$, denoted $A_O(b)$, is defined as:*

$$A_O(b) = \{a \in A(b) \mid b' = T(b, a),$$
$$\exists p, p' \in O(a, b'), p \neq p'\}$$

**Definition 3.** *The number of history-based sensor actions in execution $h$, denoted $\Psi_O(h)$, is defined as:*

$$\Psi_O(h) = |\{i = 1, 2, \cdots \mid a_i \in A_O(b_i)\}|$$

*where $h = (a_1, p_1, a_2, p_2, \dots)$.*

**Definition 4.** *A history-based contingent plan $\pi_h$ is a history-based $k$-branching-bounded contingent plan with $k \in \mathbb{N} \cup \{\infty\}$ iff $\pi_h$ satisfies:*

$$\max_{h \in H^*} \Psi_O(h) \leq k$$

*where $H^*$ is the set of complete executions of $\pi_h$.*

Definition 4 says that a history-based $k$-branching-bounded contingent plan is a history-based contingent plan where, in the corresponding history-based policy graph, there is at most $k$ branch points on any path from the root node to a leaf node (e.g. as in Figure 3). If $k = 0$ (resp. $k = \infty$), then a history-based $k$-branching-bounded contingent plan is a conformant plan (resp. contingent plan). This definition is similar to the definition of balanced $k$-contingency plans from (Meuleau and Smith 2002). As we will see in subsequent sections, however, this definition is too strong in the context of a special type of history-based policy known as a belief-based policy.

### 3.2 Belief-Based Policies

A history-based policy $\pi_h$ is called a belief-based policy if $\pi_h(h_i) = \pi_h(h_j)$ for any $h_i, h_j \in H'$ such that $b_i = b_j$ and $|h_i| = |h_j|$. For this reason, a belief-based policy can

---

[5] An equivalent definition is a rooted acyclic graph over histories where each edge is labelled with a percept.

be defined as a function $\pi : X \to A$ where $X \subseteq B \times D$ with $D \subseteq \mathbb{N}$ the set of time steps. We say that $\pi$ is stationary if $\pi(b,t) = \pi(b,t')$ for all $t, t' \in D$ such that $t \neq t'$, otherwise $\pi$ is non-stationary. A stationary belief-based policy can be defined as a function $\pi : B' \to A$ where $B' \subseteq B$. Belief-based policies are typically easier to find than their history-based counterparts: the belief space is large but bounded, whereas the history space is unbounded. Analogous to history-based policy graphs, a belief-based policy can be encoded as a rooted multigraph over (time-indexed) belief states, called a belief-based policy graph. Importantly, while history-based policies lead to policy graphs that are trees, belief-based policy graphs can be more compact, since it is possible to arrive at the same node via different executions. In fact, if the policy is stationary, then a belief-based policy graph may exhibit cycles, because it is also possible to return to a previously visited node.

The notion of a strong solution for (acyclic) history-based policies is extended to (potentially cyclic) belief-based policies through the notion of a *strong-cyclic* solution (Cimatti et al. 2003). Intuitively, cycles in a belief-based policy can lead to infinite executions, but such executions are only permitted when they are *unfair*. Formally, an infinite execution $h$ is said to be fair if, when action $a$ is executed an infinite number of times in belief state $b$, then every percept $p \in O(a, b')$ with $b' = T(b,a)$ is also observed an infinite number of times, otherwise $h$ is unfair. A belief-based policy $\pi$ is said to be a strong-cyclic solution to a contingent planning task $(M, b_1, S_G)$ if, for each complete execution $h_i$, either: (i) $h_i$ is finite and $b_i \subseteq S_G$, or (ii) $h_i$ is infinite and unfair. It follows that a strong solution is a strong-cyclic solution where every complete execution is finite.

**Definition 5.** *A belief-based contingent plan is a belief-based policy $\pi$ that is a strong-cyclic solution to a contingent planning task $(M, b_1, S_G)$.*

In the context of history-based policies, branching actions are those action-executions that can lead to distinct successor histories (i.e. distinct nodes in the policy graph). However, the fact that it is possible to arrive at the same (time-indexed) belief state via different executions suggests that Definition 2 is not valid in the context of belief-based policy graphs. Thus, in order to better understand branching in belief-based policies, let us now explore the relationship between possible percepts and successor belief states:

**Lemma 1.** $1 \leq |\{U(a,b,p) \mid p \in O(a,b)\}| \leq |O(a,b)|$.

*Proof.* By definition, $O(a,b) \subseteq P$ such that $O(a,b) \neq \emptyset$. Thus, $|O(a,b)| \geq 1$. Similarly, if $p \in O(a,b)$, then $U(a,b,p) \in B$, otherwise $U(a,b,p) =$ *undefined*. Thus, $|\{U(a,b,p) \mid p \in O(a,b)\}| \geq 1$ and $|\{U(a,b,p) \mid p \in O(a,b)\}| \leq |O(a,b)|$. $\square$

**Lemma 2.** *It is guaranteed that $|O(a,b)| = |\{U(a,b,p) \mid p \in O(a,b)\}|$ iff $O$ is deterministic.*

*Proof.* Given Lemma 1, we just need to prove (i) that there exists a bijection (i.e. a one-to-one correspondence) $f :$

$O(a,b) \to \{U(a,b,p) \mid p \in O(a,b)\}$ when $O$ is deterministic, and (ii) that such a bijection is not guaranteed to exist when $O$ is non-deterministic.

**(i)** Suppose $O$ is deterministic. By Definition, $O(a,s)$ is a singleton for each $s \in b$. Similarly, if $s \in b$ and $p \in O(a,s)$, then $s \in U(a,b,p)$. Conversely, if $s \in b$ and $p \notin O(a,s)$, then $s \notin U(a,b,p)$. It follows that, if $O(a,b) = \{p_1, \ldots, p_n\}$, then $\{f(p_1), \ldots, f(p_n)\}$ forms a partition[6] of $b$ with $f(p_i) = U(a,b,p_i)$, which satisfies the definition of a bijection.

**(ii)** Suppose $O$ is non-deterministic such that $O(a,s) = P$ for each $a \in A$ and each $s \in b$ with $|P| > 1$. By definition, we have that $O(a,b) = P$. Moreover, $U(a,b,p) = b$ for each $p \in P$, since $s \in U(a,b,p)$ iff $s \in b$. It follows that $|O(a,b)| > |\{U(a,b,p) \mid p \in O(a,b)\}| \Leftrightarrow |P| > |\{b\}|$, which contradicts the definition of a bijection. $\square$

**Corollary 1.** $U(a,b,p) = b$ *if* $O(a,b) = \{p\}$.

*Proof.* This follows from proof (i) of Lemma 2 and the fact that $O(a,b) = \{p\}$ is a deterministic observation, regardless whether $O$ is itself a deterministic function. $\square$

Lemma 1 says, firstly, that every action-execution is guaranteed to result in at least one possible percept and one successor belief state and, secondly, that the number of possible percepts is an upper bound on the number of possible successor belief states. Lemma 2 then says that, if $O$ is deterministic, there exists a unique successor belief state for each possible percept, but that this is not guaranteed if $O$ is non-deterministic. Finally, Corollary 1 says that, if there is only one possible percept (whether $p = \varnothing_P$ or otherwise), then the (single) successor belief state will be the same as the predicted belief state. Given these properties, we can now propose a definition of branching actions in the context belief-based policies that remains valid for both deterministic and non-deterministic observations:

**Definition 6.** *The set of belief-based sensor actions in belief state $b \in B$, denoted $A_U(b)$, is defined as:*

$$A_U(b) = \{a \in A(b) \mid b' = T(b,a), \exists p, p' \in O(a,b'),$$
$$U(a,b',p) \neq U(a,b',p')\}$$

**Proposition 1.** $A_U(b) \subseteq A_O(b)$.

*Proof.* By definition, $a \in A_O(b)$ iff $O(a,b')$ is not a singleton with $b' = T(b,a)$. Conversely, $a \in A_U(b)$ iff $\{U(a,b,p) \mid p \in O(a,b')\}$ is not a singleton. Thus, it follows from Lemma 1 that, if $a \in A_U(b)$, then it must also be that $a \in A_O(b)$. $\square$

**Proposition 2.** *It is guaranteed that $A_U(b) = A_O(b)$ iff $O$ is deterministic.*

*Proof.* This follows directly from Lemma 2 and Proposition 1, in that a bijection $f : O(a,b) \to \{U(a,b,p) \mid p \in O(a,b)\}$ is guaranteed to exist iff $O$ is deterministic. $\square$

---

[6] This observation has been made previously (Russell and Norvig 2009, Chapter 4).

**Definition 7.** *The number of belief-based sensor actions in execution h, denoted $\Psi_U(h)$, is defined as:*

$$\Psi_U(h) = |\{i = 1, 2, \cdots \mid a_i \in A_U(b_i)\}|$$

*where $h = (a_1, p_1, a_2, p_2, \dots)$.*

**Definition 8.** *A belief-based contingent plan $\pi$ is a belief-based k-branching-bounded contingent plan with $k \in \mathbb{N} \cup \{\infty\}$ iff $\pi$ satisfies:*

$$\max_{h \in H^*} \Psi_U(h) \leq k$$

*where $H^*$ is the set of complete executions of $\pi$.*

The problem with Definition 8 is that, in the context of bounded branching, a (time-indexed) belief state is not a *sufficient statistic* (Striebel 1965) for a history. Specifically, we know it is possible to arrive at the same (time-indexed) belief state via different executions, but this also means that those executions may contain different numbers of sensor actions; this may have implications for the choice of action, and may even preclude further sensing. For example, suppose there are two executions $h_i$ and $h_j$ such that $b_i = b_j$ and $|h_i| = |h_j|$. If $\Psi(h_i) = k - 1$, then the best action (to reach the goal) in $h_i$ might be to execute a sensor action, but if $\Psi(h_j) = k$, then executing a sensor action in $h_j$ is not an option, although the goal may still be reachable from $b_j$ via some other non-sensor action. This suggests that belief-based policies (whether stationary or not) are too restrictive to properly capture bounded branching. In the next section, we will solve this problem by proposing a generalisation of belief-based policies, called tracking-based policies.

## 3.3 Tracking-Based Policies

A history-based policy $\pi_h$ is called a tracking-based policy with $k \in \mathbb{N} \cup \{\infty\}$ if $\pi_h$ satisfies the following: (i) $h \notin H'$ if $\Psi_U(h) > k$; (ii) $\pi_h(h_i) \notin A_U(b_i)$ if $\Psi_U(h_i) = k$; and (iii) $\pi_h(h_i) = \pi_h(h_j)$ for any $h_i, h_j \in H'$ such that $b_i = b_j$ and $k - \Psi_U(h_i) = k - \Psi_U(h_j)$. As such, a tracking-based policy can be defined as a function $\pi_k : X \to A$ where $X \subseteq B \times D$ with $D = \{t \in \mathbb{N} \mid t < k\} \cup \{k\}$ the set of decision steps. Intuitively, a tracking-based policy generalises a belief-based policy where $\pi_k(b, t)$ denotes the action to execute in belief state $b$ with $t$ remaining sensor actions.[7] This is similar to belief-based policies in fault-tolerant planning, where actions may depend on the number of "failures so far" (Domshlak 2013). Conversely, while non-stationary belief-based policies are typical in finite horizon planning problems (Geffner and Bonet 2013, Chapter 6), it is worth emphasising that branching-bounded contingent planning problems do not technically have a finite horizon (e.g. if $k = \infty$, or if the domain is non-observable).

**Definition 9.** *A tracking-based contingent plan is a tracking-based policy $\pi_h$ that is a strong-cyclic solution to a contingent planning task $(M, b_1, S_G)$.*

As with belief-based policies, a tracking-based policy can be encoded as a rooted multigraph over (step-indexed) belief states, called a tracking-based policy graph. As such, tracking-based policy graphs are generally more compact than history-based policy graphs, and may contain cycles.

---

[7]Time-dependent tracking-based policies are also possible.

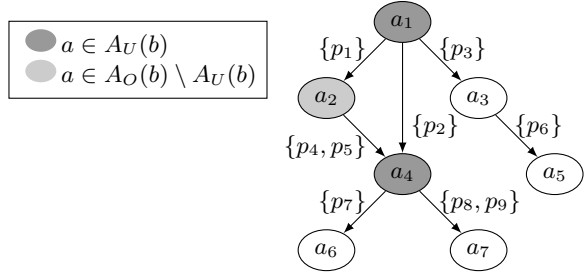

Figure 4: Tracking-based $k$-branching-bounded plan, $k \geq 2$.

**Proposition 3.** *Let $\pi_k$ be a tracking-based contingent plan with $H^*$ its set of complete executions. Then $\pi_k$ satisfies:*

$$\max_{h \in H^*} \Psi_U(h) \leq k$$

*Proof.* By definition, if $\pi_k$ is a tracking-based policy and $\Psi_U(h) > k$, then $h \notin H'$. If $h \notin H'$, then by definition $(h, a, p)$ cannot be a possible execution of $\pi_k$ for any $a \in A$ and any $p \in P$. Conversely, if $\Psi_U(h_i) = k$ and $h_i \in H'$, then by definition $(h_i, a, p)$ cannot be a possible execution of $\pi_k$ with $a = \pi_k(b_i, 0)$ for any $p \in P$ if $a \in A_U(b_i)$. Finally, if $h$ is not a possible execution of $\pi_k$, then by definition $h$ cannot be a complete execution of $\pi_k$. Thus, it must be that $\Psi_U(h) \leq k$ if $h$ is a complete execution of $\pi_k$. $\square$

**Definition 10.** *A tracking-based contingent plan $\pi_k$ is also called a tracking-based k-branching-bounded contingent plan.*

Proposition 3 says that tracking-based contingent plans directly encode the intuition of belief-based $k$-branching-bounded contingent plans. Once again, Definition 10 is similar to balanced $k$-contingency plans from (Meuleau and Smith 2002), and if $k = 0$ (resp. $k = \infty$), then a tracking-based $k$-branching-bounded contingent plan is a conformant plan (resp. contingent plan). Finally, Figure 4 demonstrates how tracking-based contingent plans can be less sensitive to bounded branching than their history-based counterparts (e.g. the plan is a history-based $k$-branching-bounded contingent plan iff $k \geq 3$).

**Theorem 1.** *It is guaranteed that a tracking-based k-branching-bounded contingent plan $\pi_k$ is a strong solution iff $k < \infty$.*

*Proof.* We need to prove: (i) that a strong-cyclic solution may not be a strong solution when $k = \infty$, and (ii) that every strong-cyclic solution is also a strong solution if $k < \infty$.

**(i)** Suppose $k = \infty$. By definition, $\pi_k$ reduces to a stationary belief-based policy $\pi$ where $\pi(b) = \pi_k(b, \infty)$ for any $(b, \infty) \in X$. Thus, $\pi_k$ may not be a strong solution, since this is true of any stationary belief-based policy.

**(ii)** Suppose $k < \infty$. By definition, if $\pi_k(b, t) \in A_U(b)$ for some $(b, t) \in X$, then belief state $b$ can only be revisited as part of a distinct step-indexed belief state $(b, t')$ such that $t' < t$. Conversely, if $\pi_k(b, t) \notin A_U(b)$ for some $(b, t) \in X$, then $(b, t)$ can only be revisited as part of

an infinite loop with a deterministic sequence of successor belief states (i.e. no branching) leading back to $(b, t)$. Infinite loops correspond to fair executions. Thus, there cannot be an unfair infinite execution of $\pi_k$, which means that all complete executions of $\pi_k$ must be finite. $\qquad\square$

Theorem 1 guarantees that, if $k < \infty$, then a tracking-based $k$-branching-bounded contingent plan will be acyclic. Combined with our observation that (time-indexed) belief states are not a sufficient statistic for bounded branching, this theorem solves an open question about defining branching-bounded contingent plans in the context of (potentially cyclic) belief-based policies (Bonet 2010). More importantly, this theorem implies that certain types of algorithms (i.e. those for acyclic plans) may be more convenient than others for branching-bounded contingent planning.

## 3.4 Algorithm

In this section, we propose a variant of the AO* optimal top-down heuristic search algorithm for acyclic AND/OR graphs (Nilsson 1971, Chapter 3; Martelli and Montanari 1973). AO* itself is the search algorithm employed by numerous existing contingent planners (Bonet and Geffner 2000; Hoffmann and Brafman 2005; Bryce, Kambhampati, and Smith 2006), where it is typically used to construct *optimal* history-based policies incrementally (Geffner and Bonet 2013, Chapter 5). However, AO* can also be used to construct optimal *acyclic* belief-based policies via belief space search. In a similar way, our algorithm (called Bounded AO*, or BAO* for short) is able to find optimal history- and tracking-based $k$-branching-bounded contingent plans. BAO* is sound and complete when $k < \infty$, but may be incomplete when $k = \infty$ where strong-cyclic solutions are not guaranteed to be strong solutions (see Theorem 1).

We first need to formalise what we mean by optimality. Let $\Pi$ be the set of history-based policies. The cost function $C$ is extended to $\Pi$ as a function $\bar{C} : \Pi \to \mathbb{R}^+$ defined as:

$$C(\pi_h) = \begin{cases} \max\limits_{h \in H^*} \sum\limits_{i=1}^{|h|} C(a_i) & \text{if } H^* \neq \emptyset \\ \infty & \text{otherwise} \end{cases}$$

where $h = (a_1, p_1, a_2, p_2, \dots)$ and $H^*$ is the set of complete executions of $\pi_h$. Note that $C(\pi_h) = \infty$ if $\pi_h$ has an infinite execution. Of course, if $\pi_h$ is a tracking-based $k$-branching-bounded contingent plan with $k < \infty$, then Theorem 1 guarantees that all executions will be finite.

**Definition 11.** *Let $\Pi' \subseteq \Pi$ be the set of history- (resp. tracking-based) $k$-branching-bounded contingent plans. The set of optimal history- (resp. tracking-based) $k$-branching-bounded contingent plans $\Pi^* \subseteq \Pi'$ is defined as:*

$$\Pi^* = \operatorname*{argmin}_{\pi_h \in \Pi'} C(\pi_h)$$

Definition 11 says that an optimal $k$-branching-bounded contingent plan minimizes cost in the worst case (that is, the maximum cost for any complete execution). Next, we can formalise the search space of BAO* as follows:

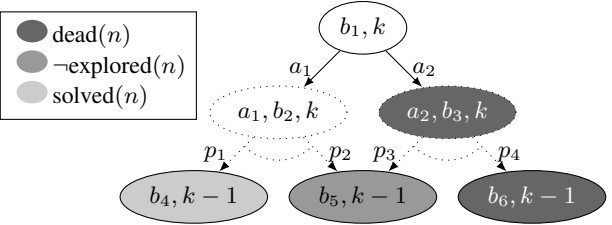

Figure 5: BAO* search graph.

**Definition 12.** *A belief space search graph is an acyclic AND/OR graph $(N_\vee \cup N_\wedge, A \cup P, E_\vee \cup E_\wedge, n)$ where $N_\vee \subseteq X$, $N_\wedge \subseteq A \times X$, $E_\vee \subseteq N_\vee \times A \times N_\wedge$, $E_\wedge \subseteq N_\wedge \times P \times N_\vee$, and $n \in N_\vee$ is the root node.*

Intuitively, nodes in a belief space search graph are (step-indexed) belief states such that OR-edges link belief states via actions, and AND-edges link belief states via percepts (see Figure 5). The step-index in each node provides the mechanism by which we track the number of sensor actions on a given path. Notice also that AND-nodes are further augmented with an action. The reason for this follows from the fact that the set of possible percepts for a given belief state depends on the action that lead to that belief state, and thus it is necessary to track those actions.

Before describing BAO* in detail, let us introduce the notion of a heuristic function in the context of BAO*. Let $V : N_\vee \cup N_\wedge \to \mathbb{R}^+ \cup \{\infty\}$ be a heuristic function. We say that $V$ is admissible if it never overestimates the cost (with respect to cost function $C$) of reaching the goal. A binary relation over nodes, denoted $\preceq_V$, is defined for nodes $n, n' \in N_\vee \cup N_\wedge$ as follows:

$$n \preceq_V n' \Leftrightarrow V(n) \leq V(n')$$

Moreover, $n \simeq_V n'$ if $n \preceq_V n'$ and $n' \preceq_V n$. Also, $n \prec_V n'$ if $n \preceq_V n'$ and $n' \npreceq_V n$. Finally, $\min(N, \preceq_V)$ denotes the single most preferred node in $N \subseteq N_\vee \cup N_\wedge$ with respect to $V$, with ties broken arbitrarily. As input, BAO* takes a contingent planning task $(M, b_1, S_G)$, an admissible heuristic function $V^*$, and a bound $k \in \mathbb{N} \cup \{\infty\}$. Importantly, the admissible heuristic function $V^*$ should satisfy the following: $V^*(b, t) = V^*(b, t')$ for all $t, t' \in D$; $V^*(a, b, t) = V^*(b, t)$ for each $a \in A$; and $V^*(a, b, t) = V^*(a', b, t')$ for all $a, a' \in A$ and all $t, t' \in D$. In other words, $V^*$ is independent of the step-index and action.

An outline of BAO* is provided in Algorithm 1, and supplementary definitions (which are identical to AO*) are provided in Table 1. In particular, Table 1b describes how another heuristic function $V$ is derived from the cost function $C$ and the input heuristic function $V^*$. The heuristic function $V$ represents a revised cost estimate and is computed by BAO* during the search. Therefore, $V$ is the heuristic function that actually guides the search, and is admissible if $V^*$ is admissible. Of course, AO* does not typically compute $V$ at each step; instead it maintains a single heuristic value for each node, which it then updates during the search via a back-propagation procedure. We omit these details for conciseness. The main changes to AO* can be found in Algorithm 1, and relate to the tracking of sensor actions on a

**Algorithm 1:** BAO*

**Input:** Contingent planning task $(M, b_1, S_G)$, admissible heuristic $V^*$, bound $k \in \mathbb{N} \cup \{\infty\}$
**Output:** $\pi_k = \text{extract}(root)$

```
1  root ← (b₁, k)
2  while ¬solved(root) ∧ V(root) ≠ ∞ do
3  │   leaf ← choose(root)
4  │   expand(leaf)
5  return extract(root)
6  function choose(n)                          /* OR-node */
7  │   if ¬expanded(n) then
8  │   │   return n
9  │   n' ← min({n''' | (n, a, n''') ∈ E∨}, ⪯ᵥ)
10 │   N'∨ ← {n''' | (n', p, n''') ∈ E∧, ¬solved(n''')}
11 │   n'' ← min(N'∨, ⪯ᵥ)
12 │   return choose(n'')
13 procedure expand(n)                          /* OR-node */
14 │   (b, t) ← n
15 │   for each a ∈ A(b) do
16 │   │   b' ← T(b, a)
17 │   │   n' ← (a, b', t)
18 │   │   if ¬path(n', n) then
19 │   │   │   E'∧ ← expand(n')
20 │   │   │   if E'∧ ≠ ∅ then
21 │   │   │   │   E∨ ← E∨ ∪ {(n, a, n')}
22 │   │   │   │   E∧ ← E∧ ∪ E'∧
23 function expand(n)                          /* AND-node */
24 │   (a, b, t) ← n
25 │   X ← ∅
26 │   for each p ∈ O(a, b) do
27 │   │   b' ← U(a, b, p)
28 │   │   X ← X ∪ {(p, b')}
29 │   if |{b' | (p, b') ∈ X}| > 1 then
30 │   │   if t = 0 then
31 │   │   │   return ∅
32 │   │   t ← t - 1
33 │   E'∧ ← ∅
34 │   for each (p, b') ∈ X do
35 │   │   n' ← (b', t)
36 │   │   if path(n', n) then
37 │   │   │   return ∅
38 │   │   E'∧ ← E'∧ ∪ {(n, p, n')}
39 │   return E'∧
```

| Predicate | Value |
|---|---|
| expanded($n$) | *true* after execution of expand($n$), otherwise *false* |
| path($n, n'$) | *true* if there is a path from $n$ to $n'$ in the current belief space search graph, otherwise *false* |
| goal($n$) | *true* if $n \in N_\vee$ and $b \subseteq S_G$ with $n = (i, b)$, otherwise *false* |
| dead($n$) | *true* if expanded($n$) and $n$ has no children, otherwise *false* |
| solved($n$) | *true* if goal($n$), or $n \in N_\vee$ and solved($n'$) for some child $n'$ of $n$, or $n \in N_\wedge$ and solved($n'$) for each child $n'$ of $n$, otherwise *false* |

(a) Predicates in BAO*.

| $V(n)$ | Condition |
|---|---|
| 0 | If goal($n$) |
| $\infty$ | If dead($n$) |
| $\min_{n' \in N} C(a) + V(n')$ | If $n \in N_\vee$ and expanded($n$) such that $N$ is the set of children of $n$ and $n' = (a, b)$ |
| $\max_{n' \in N} V(n')$ | If $n \in N_\wedge$ and expanded($n$) such that $N$ is the set of children of $n$ |
| $V^*(n)$ | Otherwise |

(b) Heuristic function $V$ in BAO*.

Table 1: Supplementary details for BAO*.

**Lines 29–32** We record the remaining number of sensor actions as $t' = t - 1$ if $a \in A_U(b)$, or $t' = t$ otherwise.

**Lines 33–39, 20–22** If possible to add an AND-edge from $n'$ to OR-node $n'' = (b'', t')$ without creating a cycle, then we add $n'$ as a child of $n$ and each $n''$ as a child of $n'$.

**Line 2** The search terminates when the root node is solved, or is deemed to be unsolvable via $V(n) = \infty$.

**Line 5** A tracking-based $k$-branching-bounded contingent plan is returned, if found, via extract($n$).

Notice that a negative result at line 18 or line 36 does not mean that no solution exists from $n$ involving $n'$ or $n''$, but simply that no acyclic solution exists (Russell and Norvig 2009, Chapter 4). Specifically, the admissible heuristic function $V$ in (B)AO* ensures that, if $n'$ or $n''$ are part of the optimal solution, then they will be part of the solution at the point that they were originally expanded. Of course, while Definition 12 and Algorithm 1 focus on tracking-based policies, a simpler variant (i.e. tree-based search, omitted due to space considerations) can also be used to incrementally construct history-based $k$-branching-bounded contingent plans.

# 4 Related Work

The only other practical treatment of branching-bounded contingent planning appears to be the work of Meuleau and Smith (2002), whose method is known to be valid only under full observability (Bonet 2010). Thus, our work represents the first practical treatment of this problem that

given path, as well as the avoidance of cycles. We can summarise the algorithm as follows:

**Lines 3–4** In each iteration, we select an OR-node $n = (b, t)$ for expansion in the current best partial solution.

**Lines 14–16** For each applicable action $a \in A(b)$, we generate the predicted belief state $b' = T(b, a)$.

**Lines 17–19, 24–28** If possible to add an OR-edge from $n$ to AND-node $n' = (a, b', t)$ without creating a cycle, then for each possible percept $p \in O(a, b)$ we generate the successor belief state $b'' = U(a, b', p)$.

is valid in the general case (i.e. partial observability). As far as we are aware, the only other work that deals with branching-bounded contingent plans has been theoretical analyses of computational complexity in planning (Baral, Kreinovich, and Trejo 2000; Bonet 2010). Bounded branching is a subclass of the broader problem of planning with bounded parameters. In this broader class, finite-horizon planning is perhaps the best known instance, requiring that plans have some bounded execution length (e.g. Rintanen 2007). Another example is conformant probabilistic planning, where *satisficing* plans guarantee some lower bound on the probability of goal achievement under an indefinite horizon (Domshlak and Hoffmann 2006). In fault-tolerant planning, partial plans are permitted under the assumption that only a bounded number of *non-primary* effects will occur during execution (Domshlak 2013). Recent work on compact plans requires that $\pi(\cdot)$ be defined only for some bounded number of *controller states* (Geffner and Geffner 2018) or *memory states* (Chatterjee, Chmelík, and Davies 2018; Pandey and Rintanen 2018). Of course, while all these works belong to the broad class of bounded planning problems, they do not share the same characteristics as branching-bounded contingent planning: they do not reduce branching, and do not generalise conformant planning.

Contingent planners that rely on belief space search can be classified in terms of: (i) their underlying search algorithm for AND/OR graphs; (ii) their belief state representation; and (iii) their heuristics. Our method is agnostic to (ii) and (iii). The best-known algorithm for (i) is probably AO* (Nilsson 1971, Chapter 3; Martelli and Montanari 1973), but other examples include LAO* (Hansen and Zilberstein 2001), LDFS (Bonet and Geffner 2005), and A*LD (Felzenszwalb and McAllester 2007). Contingent planners based on AO* include GPT (Bonet and Geffner 2000), Contingent-FF (Hoffmann and Brafman 2005), and POND (Bryce, Kambhampati, and Smith 2006). To the best of our knowledge, belief space search remains state-of-the-art in general[8] contingent planning (e.g. Contingent-FF can be regarded as state-of-the-art). That being said, alternative techniques include plan space search (Weld, Anderson, and Smith 1998), answer set programming (Tu, Son, and Baral 2007), and compilation (Muise, Belle, and McIlraith 2014). We expect that branching-bounded contingent planning could be achieved with many of these techniques.

## 5 Conclusion

An implementation of this work is available online.[9] From a practical perspective, we hope to further develop this implementation into a planner that is competitive with (but subsumes) existing state-of-the-art conformant and contingent planners. This could be achieved, for example, by the use of better heuristics (Bryce, Kambhampati, and Smith 2006), or compact belief state representations (Darwiche 2011). Subsequently, we hope to experimentally validate this work using benchmarks that permit different types of branching-bounded contingent plans (e.g. conformant and non-conformant plans). Finally, it is worth mentioning that this work was motivated by an XAIP application involving the recommendation of plans for execution by humans, so it would be interesting to evaluate the effect of bounded branching on human comprehension.

From a theoretical perspective, there are many interesting directions for future work. For example, we have defined branching-bounded contingent plans in terms of a *local* bound (i.e. where branching is bound only on complete paths of the policy graph), but a *global* bound may also be desirable (i.e. where branching is bound across the entire policy graph). The latter would be comparable to the notion of a general $k$-contingency plans from (Meuleau and Smith 2002). Such plans might be found by maintaining an explicit partial solution during the search, then discarding when the bound is exceeded. However, this would likely complicate backtracking. Another example is that, while we bound the number of branch points, we do not bound the number of branches (i.e. child nodes). Doing so could help to further reduce the complexity of the plan. Finding complete plans would certainly require an online partitioning of $O(a, b)$ such that successor belief states could be defined for sets of percepts in that partition, e.g. where $U(a, b, \{p_1, \ldots, p_n\}) = U(a, b, p_1) \cup \ldots \cup U(a, b, p_n)$. However, optimally choosing that partition is not straightfoward, and there would be no guarantees on the optimality of the resulting plan. As an alternative, fault-tolerant planning techniques (Domshlak 2013) could be used to find partial plans that only account for a bounded number of "most significant" branches.

## Acknowledgements

This work received funding from the European Union's Horizon 2020 research and innovation programme, through the DEVELOP project, under grant agreement No. 688127.

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
