# OpenReview forum: "Branching-Bounded Contingent Planning via Belief Space Search"
_icaps-conference.org/ICAPS/2019/Workshop/XAIP — XAIP 2019_

### Official Review · AnonReviewer2 · 2019-05-08
**Review2**

**Rating:** 3
**Confidence:** 2

**Review:**

The paper formalizes and provides a solver for branching-bounded Contingent Planning.  This problem generalizes both conformant and contingent planning by allowing sensing actions to be bounded.  If no sensing actions take place, then the problem is equal to conformant planning, while allowing unbounded sensing actions is fully contingent planning. A sweet spot somewhere in the middle is more realistic for many problems, where the agent might want to take up to k sensing actions.  This paper formalizes k-bounded contingent planning and provides an important step toward linking these two kinds of problems.

Generally, the paper is clear and easy to read.  My main concern relates to how well it fits with the workshop aims.  Although I can see how it might be used for explainable planning, the paper itself doesn't make a very strong claim.  Still, I do think the general approach may be useful to bring to the attention of folks attending the workshop.

My rating reflects the papers alignment with the workshop and _not_ the technical content, which is top notch.

---

### Official Review · AnonReviewer3 · 2019-05-13
**Nice formal work; relevance to workshop weak**

**Rating:** 2
**Confidence:** 2

**Review:**

The paper discusses branching-bounded contingent planning, a generalization of what is usually called contingent planning, where instead if binary observation actions we have a POMDP-style observatin function and can construct policies of varying forms depending on what they receive as input. The paper does a great job of formalizing and discussing the problem, and offering an algorithmic solution. There is no empirical evaluation yet but for a workshop paper I don't see this as a problem.

The main problem I do see is the relevance to this workshop. The authors say their work is motivated by explainability, based on the plausible hypothesis that understandability of a plan depends on the amount of branching. While this is fine and may be relevant to XAIP in the long term, the technical content of the paper is 100% contingent planning and belief space search. It seems doubtful that this will be interesting to the XAUP auience specifically (HSDIP may have been a better guess). Given the large number of submissions to XAIP, it is not clear whether it makes sense to include the paper.

---

### Decision · Program_Chairs · 2019-05-15

**Decision:**

Accept

**Comment:**

While the reviewers view this paper critically, in the spirit of making the workshop a venue for discussion and feedback we decided to reject only those papers with strong reject votes.

Please address the review criticism as best possible for the final paper version and its presentation at the workshop. In particular, please carefully discuss the relation to/links to XAIP, and the XAIP literature. Looking forward to discuss your work at the workshop!